# Impact of ozone and inlet design on the quantification of isoprene-derived organic nitrates by thermal dissociation cavity ring-down spectroscopy (TD-CRDS)

Patrick Dewald[1], Raphael Dörich[1], Jan Schuladen[1], Jos Lelieveld[1], and John N. Crowley[1]

[1]Atmospheric Chemistry Department, Max-Planck-Institut für Chemie, 55128 Mainz, Germany

*Correspondence to*: John N. Crowley (john.crowley@mpic.de)

**Abstract.** We present measurements of isoprene-derived organic nitrates (ISOP-NITs) generated in the reaction of isoprene with the nitrate radical ($NO_3$) in a 1 $m^3$ Teflon reaction chamber. Detection of ISOP-NITs is achieved via their thermal dissociation to nitrogen dioxide ($NO_2$), which is monitored by cavity ring-down spectroscopy (TD-CRDS). Using thermal dissociation inlets (TDIs) made of quartz, the temperature-dependent dissociation profiles (thermograms) of ISOP-NITs measured in the presence of ozone ($O_3$) are broad (350 to 700 K), which contrasts the narrower profiles previously observed for e.g. isopropyl nitrate (iPN) or peroxy acetyl nitrate (PAN) under the same conditions. The shape of the thermograms varied with the TDI's surface to volume ratio and with material of the inlet walls, providing clear evidence that ozone and quartz surfaces catalyse the dissociation of unsaturated organic nitrates leading to formation of $NO_2$ at temperatures well below 475 K, impeding the separate detection of alkyl nitrates (ANs) and peroxy nitrates (PNs). The use of a TDI consisting of a non-reactive material suppresses the conversion of isoprene-derived ANs at 473 K, thus allowing selective detection of PNs. The potential for interference by the thermolysis of nitric acid ($HNO_3$), nitrous acid (HONO) and $O_3$ is assessed.

## 1 Introduction

Understanding the atmospheric fate of nitrogen oxide (NO) and nitrogen dioxide ($NO_2$) is critical as both trace-gases have a great impact on air quality and human health (Crutzen and Lelieveld, 2001; Lelieveld et al., 2015). Ambient measurement of trace-gases that function as $NO_x$ reservoirs or sinks (where $NO_X = NO + NO_2$) are thus needed to provide insight into $NO_x$ removal and transport. Organic compounds with nitrate functionality can serve as $NO_X$ reservoirs in the troposphere (Thornton et al., 2002; Horowitz et al., 2007) and are generally categorised as peroxy nitrates (PNs, $RO_2NO_2$, with peroxy acetyl nitric anhydride (PAN) being its most abundant representative in the troposphere) and alkyl (aliphatic) nitrates (ANs, $RONO_2$). PNs are formed via the reaction of organic peroxy radicals ($RO_2$) with $NO_2$ (R1); ANs are formed via the minor (termolecular) channel of the reaction of $RO_2$ with NO (R2a). The competitive bimolecular process leads to alkoxy radicals (RO, R2b). During the daytime, $RO_2$ is formed mainly by the oxidation of volatile organic compounds (VOCs) by hydroxyl radicals (OH) in air (R3), with ozonolysis important at night (R4):

$$RO_2 + NO_2 + M \quad \rightarrow \quad RO_2NO_2 + M \tag{R1}$$

| 30 | $RO_2 + NO + M$ | $\rightarrow$ | $RONO_2 + M$ | (R2a) |
|----|----------------|---------------|--------------|-------|
| | $RO_2 + NO$ | $\rightarrow$ | $RO + NO_2$ | (R2b) |
| | $VOC + OH\ (+ O_2)$ | $\rightarrow$ | $RO_2 + products$ | (R3) |
| | $VOC + O_3\ (+ O_2)$ | $\rightarrow$ | $RO_2 + products$ | (R4) |

At nighttime, when OH radicals and NO are significantly less abundant, the $NO_3$ radical can initiate the oxidation of many

VOCs that contain a double-bond (Ng et al., 2017). $NO_3$, formed in the oxidation of $NO_2$ by $O_3$ (R5), is photolysed rapidly by sunlight (R6) and also reacts efficiently with NO (R7) so that it is generally of minor importance during the day (Wayne et al., 1991).

| | $NO_2 + O_3$ | $\rightarrow$ | $NO_3 + O_2$ | (R5) |
|----|----------------|---------------|--------------|-------|
| | $NO_3 + h\nu$ | $\rightarrow$ | $NO + O_2$ | (R6) |
| 40 | $NO_3 + NO$ | $\rightarrow$ | $2\ NO_2$ | (R7) |

$NO_3$ readily undergoes reactions with many unsaturated organic trace-gases of biogenic origin including isoprene, mono-terpenes and sesqui-terpenes to form organic nitrates in high yields (Ng et al., 2017; Wennberg et al., 2018; Mellouki et al., 2020). The focus of this work is the formation of organic nitrates in its reaction (in air) with the $C_5$-di-ene isoprene (ISOP, R8)

| | $NO_3 + ISOP\ (+ O_2)$ | $\rightarrow\rightarrow$ | ISOP-NIT | (R8) |
|----|----------------|---------------|--------------|-------|

where ISOP-NIT represents an isoprene-derived nitrate. Isoprene is with a total global emission of $\approx$ 500 Tg yr$^{-1}$ significantly released to the atmosphere (Guenther et al., 2012) and a large fraction of the organic nitrates formed at night-time is attributed to the reaction between $NO_3$ and isoprene (R8) (Carlton et al., 2009). The atmospheric oxidation of isoprene involving OH, $O_3$ and $NO_3$ as oxidizing agents is complex and leads to a huge variety of products (Ng et al., 2017; Wennberg et al., 2018) including multifunctional, unsaturated nitrates such as $O_2NOCH_2C(CH_3)=CHCHO$, $O_2NOCH_2C(CH_3)=CHCH_2OH$ or

$O_2NOCH_2C(CH_3)=CHCH_2OOH$ among other secondary oxidation products like dinitrates or epoxides (Wu et al., 2020).

Studies of the $NO_3$–induced oxidation of isoprene in air report AN yields between 60 and 100 % (Barnes et al., 1990; Berndt and Boge, 1997; Perring et al., 2009; Kwan et al., 2012; Schwantes et al., 2015; IUPAC, 2017; Wu et al., 2020; Brownwood et al., 2021) and the $NO_3$-induced oxidation of isoprene is responsible for a dominant fraction of organic nitrates observed in rural environments with strong biogenic emissions (Beaver et al., 2012). The major fate of isoprene-derived organic nitrates

formed in the boundary layer is deposition onto particulate matter to form nitric acid ($HNO_3$) or secondary organic aerosols (SOA) leading to largely irreversible removal of $NO_X$ from the gas phase (Ng et al., 2008; Rollins et al., 2009; Fry et al., 2018; Hamilton et al., 2021).

Individual isoprene nitrates have been measured selectively in the atmosphere by mass-spectrometric methods (Wolfe et al., 2007; Wu et al., 2020). An alternative detection scheme, in which the sum of all atmospheric PNs ($\Sigma$PNs) and ANs ($\Sigma$ANs)

are separately measured, takes advantage of their different C-N bond energy by combining thermal dissociation to $NO_2$ (R9 and R10) with detection of the latter with means of laser-induced fluorescence (TD-LIF) or cavity ring-down spectroscopy (TD-CRDS).

| RO$_2$NO$_2$ + M | $\rightarrow$ | RO$_2$ + NO$_2$ | (R9) |
|---|---|---|---|
| RONO$_2$ + M | $\rightarrow$ | RO + NO$_2$ | (R10) |

Several instruments using thermal dissociation inlets consisting of fused silica (quartz) with residence times between tens to hundreds of milliseconds have been described in the literature (Day et al., 2002; Paul et al., 2009; Wild et al., 2014; Sobanski et al., 2016; Thieser et al., 2016; Keehan et al., 2020). In these instruments, quantitative conversion of PAN is reported for temperatures between 375 and 420 K. Generally, the temperature dependence of the instrument's response to ANs has been tested using mostly saturated organic nitrates such as isopropyl nitrate (iPN) and isobutyl nitrate, which are dissociated to NO$_2$

at temperatures between 500 and 675 K. The temperatures at which PNs and ANs are quantitatively converted to NO$_2$ thus differ by ~ 200 K and are largely independent of their organic backbone (Kirchner et al., 1999; Wild et al., 2014) allowing separate measurement of the sum of all alkyl nitrates (ΣANs) and of the sum of all peroxy nitrates (ΣPNs). These observations have provided the basis for analysis of field data in which an unknown mixture of PNs and ANs are present. We note however, that most of the first generation ANs formed in the NO$_3$ + isoprene system still contain a double-bond (Barnes et al., 1990;

Skov et al., 1992; Schwantes et al., 2015) which renders them more reactive towards oxidizing agents than e.g. iPN. A well characterised thermogram for aliphatic nitrates derived from the oxidation of e.g. isoprene is thus a pre-requirement for extracting the mixing ratios of PNs and ANs from ambient measurements when using a TD-inlet. To date, only one such thermogram has been presented (Brownwood et al., 2021) which appears to be the result of a single experiment (i.e. no variation of experimental conditions) using a sample that was not stable over time. The thermogram also features slopes before

and after the ANs transition temperature which is consistent with the ideal behaviour of e.g. iPN.

In this study, we generated ISOP-NITs by reacting isoprene and NO$_3$ in a Teflon simulation chamber and used a custom-built, five-channel Cavity-Ring-Down Spectrometer (CRDS) (Sobanski et al., 2016) to analyse the organic nitrates formed. In the presence of O$_3$ we find that ISOP-NIT does not behave like the saturated analogue iPN in our quartz TD-inlet and we characterised the processes (both gas-phase and surface-catalysed) that lead to the observed behaviour. We also examined the

potential role of surface-catalysed dissociation of HNO$_3$ and nitrous acid (HONO) to NO$_2$ as well as the effect of humidity as a potential bias to measurements of PNs and ANs.

## 2 Experimental

### 2.1 Simulation Chamber

In order to analyse organic nitrates formed from the NO$_3$ + isoprene system under realistic operational conditions  for the 5-

Channel-CRDS (e.g. normal sample flow rates), we constructed a dynamic, flow-through simulation chamber SCHARK (Simulation CHamber for Atmospheric Reactions and Kinetics) of volume 1 m$^3$ (cubic, all sides ~1m long) made of PFA foil of 0.005 in. (~0.13 mm) thickness (Ingeniven).  The chamber is operated at ambient pressure and temperature; a magnetically coupled, Teflon coated propeller-type stirrer situated in the centre of the chamber floor ensures continuous mixing of the air. The trace-gas inlets and sampling ports were located at opposite corners of the cubic chamber to reduce the potential of

sampling gas that had not yet mixed. The PFA foil is surrounded by a $120 \times 120 \times 120$ cm cube constructed of four Perspex and two steel walls, the interspace ($0.7$ m$^3$) is permanently flushed with 1 SLPM (L (STP) min$^{-1}$) of dry synthetic "zero-air" in order to avoid contamination through permeation of trace-gases present in the laboratory air. The Perspex walls serve as observation windows and were covered with light-tight material during the experiments described here.

Zero air was provided by passing pressurized air through a commercial air purifier (CAP 180, Fuhr GmbH). Humidification of the air was achieved with a permeation source (MH-110-24-F-4, Perma Pure LLC) filled with deionized water. Typical total flow rates of 15 or 23 SLPM zero air into the chamber result in exchange rates $k_{exch}$ of 2.7 or 4.2 x 10$^{-4}$ s$^{-1}$, i.e. lifetimes of gases in the chamber of ~ 40-60 minutes. Note that in "flow-through" operation, the concentrations of trace gases in the chamber are controlled both by chemical processes and by the rate of flow into (and out of) the chamber so that "steady-state" is achieved on the order of hours.

Ozone mixing ratios in the SCHARK were measured by sampling 2 SLPM through a ~3 m long section of 0.25 in. (outer diameter, OD) PFA-tubing to a commercial ozone monitor (2B Technologies, model 205) with a detection limit of ~ 1 part per billion by volume (ppbv) and 5 % uncertainty. $O_3$ measurements were also used to establish the time required (under standard flow conditions) to achieve complete mixing within the chamber (< 1 minute) and to derive the exchange rate by monitoring the exponential rise or decay of $O_3$ when its supply was switched on or off (Fig. S1 in the Supplement). $O_3$ (up to 600 ppbv) was generated by passing a fraction of the air flowing into the chamber through a UV-transparent cuvette (~ 70 cm³) illuminated by a low-pressure Hg-lamp (PenRay) that dissociated $O_2$ (to O atoms and thus $O_3$) at 185 nm.

. A known flow of isoprene entered the chamber as a dilute sample from a 12 L stainless steel storage canister (Landefeld GmbH) which was prepared manometrically from evaporation of pure isoprene (Acros Organics, 98 %) and mixing with helium (5.0, Westfalen). The isoprene concentration in the storage canister was quantified indirectly by measuring the $NO_3$ reactivity via flowtube-CRDS (Liebmann et al., 2017; Dewald et al., 2020) and was found to be 46.5 ppmv, in agreement (within 15 %) with the manometrically derived mixing ratio. A gas sample of isopropyl nitrate (Sigma Aldrich, 58 ppmv in $N_2$ 5.0, Westfalen) was prepared in a similar fashion.

Two methods of in-situ $NO_3$ generation were employed. In the first, $NO_3$ was produced in the chamber via the reaction of $NO_2$ with $O_3$ (R5), whereby $O_3$ was generated as described above and $NO_2$ was taken from a bottled sample (Air Liquide, 1 ppmv in $N_2$). Typical concentrations of $NO_2$ and $O_3$ were 6-10 and 100-160 ppbv, respectively. Alternatively, $NO_3$ was generated in the thermal decomposition of $N_2O_5$ (R11) which was eluted into the chamber by passing a regulated flow of $N_2$ over $N_2O_5$ crystals held at temperatures between – 78 and –70°C.

$$N_2O_5 + M \qquad \rightarrow \qquad NO_3 + NO_2 + M \qquad\qquad\qquad\qquad (R11)$$

$N_2O_5$ was synthesised by the sequential, gas-phase oxidation of NO (5% in $N_2$, Westfalen) in an excess of $O_3$ (Davidson et al., 1978) and trapped at -78°C (acetone and dry ice). In this case, $O_3$ was obtained by electrical discharge through oxygen (5.0, Westfalen) using a commercial generator (Ozomat Com, Anseros). Note that, the latter method enables us to generate $NO_3$ in the chamber in an $O_3$-free environment.

## 2.2 Detection of organic nitrates by cavity ring-down spectroscopy (CRDS)

Simultaneous measurements of the mixing ratios of $NO_2$, $NO_3$, $N_2O_5$, $\Sigma ANs$ and $\Sigma PNs$ in the SCHARK chamber were made using a five-channel cavity ring-down spectrometer (CRDS) that has been described in detail (Sobanski et al., 2016) and only a brief summary of key features of the instrument are given here. Each of the five cavities consists of FEP-coated (FEPD 121, DuPont) stainless steel tubes which are equipped with two high-reflectivity mirrors (see below) supported 90 cm apart ($L$). The volumes in front of the mirrors are purged with dry synthetic air, which results in a reduction of the effective optical path length from 90 cm to 62.1 cm ($d$). The standard expression Eq. (1) is used to derive in-cavity concentrations [X] from the difference in ring-down constant in the absence ($k_0$) and presence ($k$) of an absorber X:

$$[X] = \frac{L}{d} \cdot \frac{1}{c\sigma_{eff}} \cdot (k - k_0) \tag{1}$$

where $c$ is the speed of light and $\sigma_{eff}$ is the effective cross-section derived from the overlap of the laser emission and the $NO_2$ (Vandaele et al., 1998) or $NO_3$ absorption spectrum (Orphal et al., 2003).

Three of the cavities are operated at 409 nm for detection of $NO_2$ whereby 409 nm light is provided by a square-wave-modulated (2500 Hz) laser-diode. The three 409 nm cavities, thermostated to 303 K and typically operated at a pressure of ~ 733 hPa, sampled from the SCHARK at a total flow rate of 6 SLPM, which initially passes through a 2.3 m long PFA inlet (1.5 m with OD 0.25 inch. and 0.8 m with OD 0.125 inch) before being split into three equal flows. One flow is directed to a cavity via an unheated, 60 cm long PFA tube (0.375 inch OD) to measure $NO_2$. The other two flows are directed through thermal dissociation inlets (TDIs) in which PNs and ANs are converted to $NO_2$. At the given conditions (i.e. flow rate, pressure, residence time in the heated section), keeping our TDI at temperatures close to 448 K, results in quantitative conversion of PNs to $NO_2$ so that the cavity sampling via this inlet measures the sum of PNs + $NO_2$. Heating our second TDI to $\approx$ 650 K results in the complete conversion of ANs to $NO_2$ so that the sum of ANs + PNs + $NO_2$ can be measured as described in the literature cited in the introduction. The choice of material for these TD-inlets has a profound influence on the results obtained, as described below.

The standard deviation ($2\sigma$) of consecutive baseline measurements define the limits of detection (LOD) which are 38, 44 and 90 pptv for [$NO_2$], [$\Sigma PNs$] and [$\Sigma ANs$] respectively under laboratory conditions. The total uncertainty for the $NO_2$ measurement is 9 % which includes uncertainty in the (effective) $NO_2$ cross-sections. For the measurements of ANs and PNs the associated uncertainties are highly dependent on the concentrations of other trace gases and the corrective procedure accounting for radical recombination effects (Sobanski et al., 2016).

For simplicity, we refer to the three cavities as the "$NO_2$ cavity" (room-temperature inlet), the "PNs cavity" (TD-inlet at circa 473 K in which $\Sigma PNs + NO_2$ are measured) and the "ANs cavity" (TD-inlet at circa 673 K in which $\Sigma ANs + \Sigma PNs + NO_2$ are measured).

The remaining two cavities of the CRDS were operated at 662 nm (laser modulation at 625 Hz) for detection of $NO_3$. While one cavity is thermostated to 303 K (and detects $NO_3$ only), the second one (as well as an FEP-coated glass reactor located upstream) is thermostated to 373 K so that $N_2O_5$ is stoichiometrically converted $NO_3$ and the summed mixing ratio of $NO_3$ and

$N_2O_5$ is obtained.  The two 662 nm cavities sampled air from the SCHARK at a total flow rate of 15 SLPM through a ~1.5 m ¼ in. (OD) PFA tube. Corrections to the mixing ratios were made to account for loss of $NO_3$ and $N_2O_5$ during transport to and through the cavities. Using the method described in Sobanski et al. (2016), $NO_3$ transmission was found to be 89 % in both cavities. The $NO_3$ and $N_2O_5$ measurements are not central to this study, but allowed the quantitative surveillance of $NO_3$ (and indirect $N_2O_5$) consumption by isoprene.

Figure 1 shows three types of thermal dissociation inlets (TDIs) used to convert organic nitrates to $NO_2$. In the original version of this instrument (Sobanski et al., 2016) the $\Sigma PNs$ and $\Sigma ANs$ cavities sampled via 12 mm ID quartz TD-inlets (TDI-1), with a length of 55 cm, the first ~10 cm of which was wrapped with heating wire. In order to reduce bias caused e.g. by the reformation of the organic nitrate after its thermal dissociation, this section was filled with glass beads (Sigma-Aldrich G9268, $\emptyset$ ~ 0.5 mm) to provide a surface for heterogeneous loss of radicals. The glass beads were supported on a 2 cm thick glass frit and reduce the pressure downstream by $\approx$ 28 hPa compared to TDI-1. Problems associated with temperature-dependent flow resistance through these small beads and the need for an extra filter (upstream) to prevent their transport into the inlet lines when flows were temporarily reversed (e.g. during instrument shut-down), led us to switch to larger beads (Merck, $\emptyset$ = 3 mm) and these were used throughout this study in TDI-1. TDI-2 is made of a quartz glass tube with the same dimensions as TDI-1, but features a longer heated section (20 cm) and is free of additional surfaces like glass beads or frits. TDI-2 is thus similar to many other thermal dissociation inlets described in the literature (see above). TDI-3 is constructed from a 55 cm long PFA tube (0.375 inch OD), where the first 20 cm are heated. The melting point of PFA is lower than the temperature required to thermally dissociate ANs, so TDI-3 could only be used for the measurement of PNs + $NO_2$. The temperature of the external wall of the TD-inlets were measured with a K-type thermocouple situated at the centre of the heated section, which was insulated with mineral wool. At a flow rate of 2 SLPM and an operating pressure of 733 mbar, approximate residence times in the inlets without glass-beads are 0.20 s (in TDI-2 at 650 K) and 0.13 s (TDI-3 at 450 K) when assuming a homogeneous temperature distribution equal to that measured on the outer wall of the tubing.

## 3 Results and Discussion

Figure 2 shows the result of an experiment in which 150 sccm $NO_2$ (1 ppmv in air) was flowed into the SCHARK along with isoprene (7 sccm of 46.5 ppmv in He) and 24 SLPM zero air of which 5 SLPM were passed over the low-pressure Hg-lamp zero-air to generate $O_3$ (~ 96 ppbv). $NO_2$ was sampled (as usual) via the room-temperature PFA inlet, the $\Sigma PNs$ (473 K) and $\Sigma ANs$ cavities (673 K) both sampled via TDI-1 (quartz tube with glass-beads). $O_3$ was added at 09:30 and $NO_2$ at 10:00 (all times are local times, LT).

Just prior to the addition of isoprene at 12:00 LT, the system is close to steady-state with ~ 5 ppbv $NO_2$ and 92 ppbv $O_3$. After subtraction of the measured $N_2O_5$ mixing ratios, a residual signal of ~ 100 pptv is detected in both PNs and ANs channel, which may be caused, as discussed below, by interference of $HNO_3$ in the ANs channel and a memory effect of the glass beads in the PNs channel (section 3.2). Note that after addition of isoprene, both $NO_3$ and $N_2O_5$ are reduced drastically ($NO_3$ ~3 pptv,

N$_2$O$_5$ ~ 5 pptv) and the thermal dissociation of N$_2$O$_5$ no longer contributes to NO$_2$ signals in the PNs and ANs channels (Sobanski et al., 2016; Thieser et al., 2016).

At 14:00 LT, the cavity sampling from the 673 K TD-inlet indicated ~ 610 pptv for the summed mixing ratio of (ΣANs + ΣPNs), whereas the cavity sampling from the 473 K TD-inlet (ΣPNs) indicated ~ 400 pptv. Since the signal in the ΣANs channel includes both the contribution of peroxy and alkyl nitrates this implies that only 210 pptv (34 %) of the detected products can be attributed to alkyl nitrates, which is inconsistent with the high yields (60-100 %) of ANs that result from the reaction of NO$_3$ with isoprene (Barnes et al., 1990; Berndt and Boge, 1997; Perring et al., 2009; Rollins et al., 2009; Kwan et al., 2012; Schwantes et al., 2015; IUPAC, 2017; Brownwood et al., 2021). Compared to ANs, we expect the mixing ratios of e.g. PAN, O$_2$NOCH$_2$C(CH$_3$)=CHC(O)O$_2$NO$_2$ or methacryloyl peroxynitrate (MPAN) in this system to be negligible as their precursors such as O$_2$NOCH$_2$C(CH$_3$)=CHCHO (Jenkin et al., 2015) or methacrolein (Kwok et al., 1996; Berndt and Boge, 1997; Schwantes et al., 2015) are oxidized only inefficiently in the dark. The formation of PNs only takes place once isoprene has been depleted so that secondary oxidation of the above-mentioned aldehydes by OH or NO$_3$ leading to further acyl-peroxy radicals (which form PNs) become at least competitive to the primary oxidation of isoprene. This is however never the case in the present experiments as isoprene is continuously flowed into the chamber and remains according to model calculations (see below) at a level of ≈ 11.4 ppbv. Given the high abundance of O$_3$ and isoprene in this system, ozonolysis of the latter together with the associated formation and decomposition of Criegée intermediates to acetylperoxy radicals CH$_3$C(O)O$_2$ (Nguyen et al., 2016; Vansco et al., 2020) should make PAN the most important, potential contributor to a signal in the PNs cavity. However, according to the branching ratios given in Nguyen et al. (2016), this reaction path is a minor one and CH$_3$C(O)O$_2$ (and thus PAN) should be formed in negligible amounts.

In order to identify the origin of the unexpectedly high ΣPNs signal when NO$_3$ and isoprene are mixed in the dark, thermograms of the NO$_3$ + isoprene system were recorded in an experiment where ~ 2.8 ppbv of isoprene-derived nitrates (as measured with TDI-2 at 625 K) were generated by flowing NO$_2$ (200 sccm of 1 ppmv) and isoprene (9.8 sccm of 46.5 ppmv) in 15 SLPM dry synthetic air (with 5 SLPM over the Hg lamp for generation of ~ 150 ppbv O$_3$). Similar to the experiment in Fig. 2, N$_2$O$_5$ mixing ratios are expected to be suppressed to a few pptv under these conditions so that its thermal dissociation (to NO$_2$) did not contribute to the ΣPNs and ΣANs signals. Using this chemical system, we simultaneously measured ISOP-NIT thermograms once steady-state established using TDI-1 (quartz, glass beads, 10 cm heated section) and TDI-2 (quartz, no glass beads, 20 cm heated section) both initially held at 703 K. Subsequently, both TDIs were cooled to ambient temperature over a period of ~ 1.75 h. The ΣANs signals from this experiment are plotted against the inlet temperature in Fig. 3a to generate the ISOP-NIT thermogram. This is displayed along with an isopropyl-nitrate thermogram (iPN, red data points) measured using the same inlets under the same flow-conditions but using iPN diluted to 5.5 ppbv in dry synthetic air sampled directly through a PFA line (together with 1.5 ppbv NO$_2$ impurity) to the instrument.

For iPN, we observe a well-defined onset of thermal dissociation at ~ 525 K with a plateau (maximum conversion) at ~ 650 K as reported previously for this setup (Sobanski et al., 2016). When measuring iPN, TDI-1 results in a slightly steeper thermogram than TDI-2 in the 575-650 K range, which may be related to changes in gas-flow and heat-transfer within the inlet

caused by the glass beads. Neither TD-inlet type results in dissociation to $NO_2$ at temperatures < 500 K. In contrast, the ISOP-NIT thermograms (normalised to the signal at the plateau at 625 K of TDI-2) indicate formation of $NO_2$ over a much broader range of temperatures (350-700 K).

The effect of humidifying the air was examined in an almost identical experiment conducted with $NO_2$ (150 sccm of 1 ppmv) and isoprene (7 sccm of 46.5 ppmv) in 15 SLPM  synthetic air with relative humidity (in the SCHARK) of 33.5 % at 22°C. In this case, ~ 2.3 ppbv ISOP-NIT was formed. The thermograms obtained with TDI-1 and TDI-2 under these conditions are depicted in Fig. 3b. The broad thermogram measured with TDI-1 is very similar to that obtained under dry conditions (Fig. 3a) although even at room temperature an additional $NO_2$ signal of 500 pptv is detected. Sampling via TDI-2, yields an ISOP-

NIT thermogram that has similar features to that obtained under dry conditions, although the peak at ~400 K has well defined minima on both flanks and is shifted to higher temperatures. In separate experiments, humidified synthetic air (RH = 40 %, 23°C) and $NO_2$ (10.8 ppbv) were sampled through a PFA line directly to the instrument. The thermogram (in the absence of isoprene or ISOP-NIT) using TDI-2 was recorded and can be found in the Supplement (Fig. S2) revealing that the presence of water and $NO_2$ in the inlet is sufficient to reproduce some features displayed in Fig. 3b with TDI-2. It is well known that $H_2O$

and $NO_2$ can react on surfaces to form HONO and $HNO_3$ (Pitts et al., 1984; Finlayson-Pitts et al., 2003) and their formation in the SCHARK was verified in section 3.1. In the presence of $H_2O$, the efficiency of conversion of ISOP-NIT to $NO_2$ drops to about 5% at ~ 460 K. This is much less than under dry conditions whereby 20 % conversion of ISOP-NIT was observed between 375 and 475 K (Fig. 3a). Within a framework for surface-catalysed conversion of ISOP-NIT to $NO_2$ presented below, this observation can be interpreted as arising from the competitive adsorption to the surface of nitrated hydroperoxides and

$H_2O$, i.e. $H_2O$ (which is vastly more abundant) reduces the surface coverage of the organic nitrate at the surface.

The results in Fig.2 and Fig.3 show that separate detection of ANs and PNs based on their thermal dissociation can be problematic for the $NO_3$ + isoprene system. Identifying the cause of this and and providing potential solutions to circumvent the problem is the aim of this work. To do this, we first focus on the "dry" experiment and highlight two regions of the thermograms in which large deviations from the expected behaviour are observed.


**Region I ($T$ > 648 K)**

Figure 3a indicates that, at temperatures above 648 K (shaded region I), the behaviour of the two TDIs diverges significantly: While use of TDI-1 (glass beads) results in an increase in $NO_2$ with increasing temperature, the use of TDI-2, leads to a decrease in the $NO_2$ signal in the same temperature range. The increase in $NO_2$ continues at temperatures above that required

to convert ANs to $NO_2$, which implies the presence of a $NO_2$–containing trace-gas where the $NO_2$-moiety is more strongly bound than in ANs.

In order to assess to which extent this behaviour is potentially caused by inorganic trace-gases that are not directly related to isoprene oxidation, an experiment with only $NO_2$ (2.75 ppbv) and $O_3$ (146 ppbv) in 23 SLPM dry synthetic air was performed. The steady-state concentration of $N_2O_5$ + $NO_3$ was measured as 78 pptv. The resulting thermograms using TDI-1 and TDI-2

and after subtraction of the signal from the $NO_2$ cavity (i.e. unheated inlet) are depicted in Fig. 4. No significant additional

signal is observed below 475 K (region I) in either of the inlets. In region II ($T > 675$ K) on the other hand, we observed an increase (by ~500 pptv) in the signal to at 703 K with TDI-1, whereas ~50 pptv are lost in TDI-2. In order to identify the trace-gas(es) responsible for the signals observed in the system without isoprene, an Iodide-Chemical-Ionization Mass Spectrometer (I-CIMS (Eger et al., 2019)) described in the Supplement (S8) was coupled to the experiment. As shown in the Supplement

(Fig. S3) both $HNO_3$ and nitrous acid (HONO) were observed as soon as $O_3$ and $NO_2$ were present in the chamber and their formation is enhanced in the presence of water vapour, which is a common phenomenon in Teflon chambers (Pitts et al., 1984). We also found that reversing the flows and sampling the air into the CIMS *after* passing through the TDI-1 or -2 (at 475 K) resulted in removal of the $HNO_3$. Sampling through TDI-1 also led to loss of HONO.

Various TD-CRDS and TD-LIF instruments report the detection of $HNO_3$ as $NO_2$ following thermal dissociation at

temperatures around 700 K (Day et al., 2002; Wild et al., 2014; Thieser et al., 2016). The sensitivity of the present set-up to $HNO_3$ was investigated by sampling nitric acid from a calibrated permeation source (Friedrich et al., 2020) via TDI-1 and TDI-2 simultaneously. In these experiments, 22 ppbv $HNO_3$ (with 780 pptv $NO_2$ impurity) in dry synthetic air was delivered to the TDIs along with 350 ppbv $O_3$. The $HNO_3$ mixing ratio was derived using a known permeation rate (Friedrich et al. 2021) and dilution factor. Figure 5 shows the temperature-dependent conversion efficiency of $HNO_3$ to $NO_2$ in the presence of ozone

(squares) with TDI-1 (black solid symbols) and TDI-2 (open symbols). Conversion of $HNO_3$ to $NO_2$ starts at ~550 K and increases with rising temperature. At 680 K, the conversion efficiency is 23 % for TDI-1 and 8 % for TDI-2. No significant conversion of $HNO_3$ to $NO_2$ was observed when ozone was absent (blue datapoints) and was drastically reduced when the synthetic air was humidified to RH = 55 % at room temperature (red datapoints). The effect of water vapour is consistent with previous observations on the effect of humidity (Sobanski et al., 2016; Thieser et al., 2016; Friedrich et al., 2020).

The decomposition of $HNO_3$ to $NO_2$ thus only occurs in the presence of ozone under dry conditions and its rate increases greatly at $T > 650$ K. This is consistent with the observations in Fig. 3b for TDI-1 and represents a likely explanation for the increase in signal when sampling from the SCHARK to investigate the $NO_3$ + isoprene system. The apparently more efficient (~ factor three) conversion of $HNO_3$ to $NO_2$ in TDI-1 than in TDI-2 is explained by the loss of $NO_2$ at high temperatures in TDI-2 through the reaction with O-atoms (see section 3.3). In TDI-1 this is prevented by the removal of O-atoms by the glass

beads (e.g. via scavenging or surface-catalysed recombination to $O_2$).

The ozone-assisted conversion of $HNO_3$ to $NO_2$ cannot be explained by known gas-phase processes as the reaction between $HNO_3$ and $O(^3P)$ (R13) has a low rate coefficient ($k_{13} < 3$ x $10^{-17}$ cm³molecule$^{-1}$s$^{-1}$ at 298 K (Burkholder et al., 2016)) and results mainly in the formation of OH and $NO_3$ (R13). The more efficient conversion of $HNO_3$ to $NO_2$ in TDI-1 (with glass beads) compared to TDI-2 indicates that a surface-catalysed process involving either ozone or $O(^3P)$ is involved (R14).

$HNO_3 + O(^3P) \rightarrow NO_3 + OH$ (R13)

$HNO_3 + O_3$ or $O(^3P)$ + surface $\rightarrow NO_2$ + products (R14)

Assuming that, in a Langmuir-Hinshelwood type process, the first step in the surface catalysed reaction is physi-adsorption of $HNO_3$ to the surface, the strong reduction in conversion of $HNO_3$ to $NO_2$ under humid conditions is explained by the

competitive adsorption of $HNO_3$ and $H_2O$, the latter favoured by its much larger concentrations. i.e. $H_2O$ drives $HNO_3$ from
the surface and thus protects it from surface reactions.

**Region II ($T$ = 350-475 K)**

In region II (350-475 K, shaded area in Fig.3), instead of the near-zero signal expected in the absence of significant amounts
of PNs or $N_2O_5$ we observe a monotonic increase in $NO_2$ with the temperature which is a factor of ~ 2 steeper in TDI-1 than
in TDI-2. The signal in TDI-1 at 475 K, where only PNs are expected to dissociate, is ~50 % of the maximum signal at 650 K.
There are several potential explanations for this behaviour which include: 1) the formation and detection of thermally less
stable ANs (including e.g. di-nitrates), which dissociate at lower temperatures than e.g. iPN, 2) the formation of non-acyl,
isoprene-derived peroxy-nitrates ($RO_2NO_2$) that are sufficiently long-lived to build up to appreciable concentrations in the
SCHARK, 3) chemical processes taking place in the TD-inlets that convert ISOP-NIT to $NO_2$. Scenario 1) appears unlikely as
several studies have shown that the O-N bond-strength in various alkyl-nitrates is very similar (Hao et al., 1994; Wild et al.,
2014). We also note that the formation of di-nitrates (in the absence of NO) only takes place when isoprene levels are very
low and the first-generation nitrates formed in the $NO_3$ + isoprene reaction can react with a further $NO_3$. This can be ruled out
for the present experiments in which the isoprene mixing ratio is always much larger than that of the first generation nitrates
formed, which in any case react much more slowly with $NO_3$ than does isoprene. The second explanation requires that $RO_2$
formed in the initial reaction between $NO_3$ and isoprene react with $NO_2$ to form $RO_2NO_2$. Given our experimental conditions,
we would indeed expect that the main fate of any $RO_2$ formed in the reaction between $NO_3$ and isoprene is reaction with $NO_2$,
which will dominate over self-reaction or reaction with $NO_3$, other $RO_2$ or $HO_2$. Non-acyl $RO_2NO_2$ are however generally
highly thermally unstable, with lifetimes (at room temperature) of seconds or minutes, with respect to re-dissociation to $RO_2$
+ $NO_2$.
For isoprene-derived $RO_2NO_2$ to contribute to the signal observed in region II would require that the $RO_2 – NO_2$ bond strength
be comparable to those of acyl nitrates such as PAN. The dominant 1,4-peroxy radical formed when $NO_3$ reacts with isoprene
has a nitrate group separated by two carbon atoms from the peroxy carbon. It seems unlikely that this could have a stabilising
effect on the O-N bond in $RO_2NO_2$ in the same way that an α-carbonyl group does. Indeed, chamber experiments investigating
the products of the $NO_3$ + isoprene reaction in detail (Barnes et al., 1990; Wu et al., 2020) have failed to identify neither acyl-
nor non-acyl-$RO_2NO_2$ as stable or semi-stable products formed from primary oxidation.
In support of scenario 3, sections 3.2 to 3.4 describe the evidence for chemical reactions leading to $NO_2$ formation that bypass
the thermodynamic barrier for direct $NO_2$ formation but are surface-catalysed, require the presence of $O_3$ in either the SCHARK
or in the inlet. These processes are peculiar to alkyl nitrates with a C=C double bond and thus have not been observed in TD-
inlets tested only with saturated alkyl nitrates such as the frequently used isopropyl nitrate.

## 3.2 The role of O₃

To further investigate the conversion of ISOP-NITs to $NO_2$ at low temperatures in the TD-inlets, we generated $NO_3$ via the room-temperature thermal decomposition of $N_2O_5$ thus ruling out chemical processes that were initiated or catalysed by $O_3$. In these experiments, $N_2O_5$ was transported into the SCHARK by passing a flow of 0.1 SLPM dry synthetic air over a crystalline $N_2O_5$ sample cooled to –78°C with further dilution in a 15 SLPM flow of zero-air. The combined concentration of $N_2O_5 + NO_3$ that remains after the reaction with ~ 22 ppbv isoprene (7 sccm of 46.5 ppmv) was measured as 40.5 pptv. The use of high isoprene concentrations guarantees that the thermal decomposition of $N_2O_5$ does not contribute significantly to the thermograms. Under these conditions several ppbv of ISOP-NIT were formed. A constant flow (2 SLPM) of zero air was passed over a low-pressure Hg-lamp and added between the sampling port of the SCHARK and the inlets and TD-inlets of the $NO_2$, PNs and ANs cavities. This way the $O_3$ mixing ratio in the TD-inlets could be varied without affecting the chemistry in the chamber.

Figure 6 presents the results of one such experiment in which ISOP-NIT was sampled from the SCHARK using TDI-1 and TDI-2, both initially held at 703 K with thermograms obtained by decreasing the temperature of both inlets in 25 K steps. At each temperature step (periods of 20 min), after recording the signal under $O_3$ free conditions (black squares), a low (40-54 ppbv, green triangles), medium (97-111 ppbv, blue triangles) and high (185-219 ppbv, orange circles) mixing ratio of $O_3$ was added in front of the TD-inlets. Before cooling to the next temperature, the signal without $O_3$ was measured again and agreed within 30-150 pptv to the value at the beginning of the corresponding period. To enable comparison with "ideal" behaviour, the thermogram of isopropyl nitrate (iPN, red circles) recorded from an experiment while flowing 162 ppbv $O_3$ (and 3 ppbv $NO_2$ impurity) through the SCHARK is also plotted.

Figure 6a displays a thermogram obtained when sampling ISOP-NITs from the SCHARK via TDI-1 (glass beads). In the presence of $O_3$, the thermograms are very broad with substantial $NO_2$ formation between 350 and 475 K (shaded region II) and in this sense are comparable to those displayed in Fig. 3, where $NO_3$ was obtained from the reaction of $NO_2$ with $O_3$. In region I, the effect of going from ~50 to ~200 ppb of ozone is to increase the $NO_2$ generated drastically. This is the opposite of that observed when sampling via TDI-2 and thus in agreement with the results of the experiment depicted in Fig. 3, where the increase in signal was assigned to detection of $HNO_3$. For the experiments in which $NO_3$ was generated in the room-temperature thermal dissociation of $N_2O_5$, $HNO_3$ can arise from reactions of $N_2O_5$ with moisture on the walls and is present as impurity in the $N_2O_5$ sample. As described above, the presence of glass beads has two effects which operate in the same direction in this temperature regime: The conversion of $HNO_3$ to $NO_2$ is catalysed by the surface provided by the glass-beads and at the same time the loss of $NO_2$ (via reaction with $O(^3P)$) is reduced as $O(^3P)$ is scavenged by the glass surface.

For TDI-1, the thermograms obtained without ozone (black squares) differ greatly to those in which ozone was present. Without ozone, $NO_2$ is not generated at temperatures lower than 550 K but its concentration increases rapidly at temperatures above ~ 600 K with no indication of a plateau being reached. The thermograms obtained in this $NO_3$-isoprene system using TDI-1 in the absence of ozone bears little resemblance to that of iPN. Furthermore, during periods without ozone or heat in

TDI-1, compounds of lower volatility like ISOP-NITs or HNO$_3$ appear to deposit on the glass beads and frit. This would form an explanation for a "memory effect" observed for TDI-1, whereby after exposure to HNO$_3$ or organic nitrates during unheated periods, an increase in the NO$_2$ signal followed by a slow decrease taking several hours is observed as soon as pure synthetic air and ozone were added to the flow through the heated inlet. An example of this phenomenon is shown in Fig. S4 in which (at peak signal) 60 ppbv of NO$_2$ was detected just by heating TDI-1 to 703 K in the presence of O$_3$ in synthetic air. In order to avoid bias in results caused by this effect, a cleaning procedure was adopted prior to all experiments whereby the inlet was heated to 703 K and exposed to ozone in synthetic air until a constant, low residual signal, usually between 20 and 200 pptv, was established. This memory effect seen for TDI-1 is also observed when a thermogram of ISOP-NIT (generated in a system similar to the experiment in Fig. 3) is measured by going from cold to hot temperatures (Fig. S5).

In Fig. 6b we present the results of the same experiment using TDI-2. The non-zero signal (~ 250 pptv) at temperatures between ~320 and 450 K) results from instability in the baseline. In the absence of ozone, the organic nitrates generated in the NO$_3$-initiated oxidation of isoprene follow a well-defined thermogram (black squares and solid line) between 475 and 650 K, which is very similar to the thermogram measured for iPN (red circles). The addition of ozone does not result in the formation of NO$_2$ in region II, but does induce NO$_2$ losses for temperatures above 650 K (region I). The fact that ISOP-NIT was not converted to NO$_2$ at temperatures < 475 K suggests that the observation of a large signal in Fig. 3a (region II, white squares) is linked to O$_3$-induced chemistry in the SCHARK which will be discussed in more detail in section 3.3. The loss of NO$_2$ in region I increases with increasing amounts of O$_3$ with ~35 % of the NO$_2$ formed at 625 K lost when O$_3$ was increased to ~ 200 ppbv. The same behaviour is observed in the thermogram of iPN which confirms that this process is independent of the nature of the nitrate but solely linked to thermal decomposition of O$_3$ and subsequent reactions of O($^3$P) with NO$_2$.

### 3.2.1 Thermal dissociation and gas-phase reactions of O$_3$ and O($^3$P)

An important clue to the underlying chemical process that lead to the conversion of ANs to NO$_2$ at temperatures lower than those required to break the O-N bond is the fact that the thermogram of iPN (measured with TDI-1) is not significantly affected by the presence of 163 ppbv O$_3$ whereas thermograms of ISOP-NIT, the vast majority of which are unsaturated, are greatly broadened when O$_3$ is present.

It is well known that O($^3$P) reacts rapidly (via electrophilic addition) to C=C double bonds (Leonori et al., 2015) and we thus assessed the potential impact of NO$_2$ formation via reactions of O$_3$ or O($^3$P) (formed in the thermal dissociation of O$_3$ in the TDIs) with ISOP-NIT.

The concentration of O($^3$P) in the TDIs depends on the concentration and rate of thermal decomposition of O$_3$ and thus on the gas-temperature as well as its rate of recombination with O$_2$, reactions with O$_3$, NO$_2$, isoprene, isoprene nitrates and loss to the walls:

| | | | |
|---|---|---|---|
| O$_3$ + M | $\rightarrow$ | O($^3$P) + O$_2$ + M | (R15) |
| O($^3$P) + O$_2$ + M | $\rightarrow$ | O$_3$ + M | (R16) |
| O($^3$P) + O$_3$ | $\rightarrow$ | 2 O$_2$ | (R17) |

| | | | |
|---|---|---|---|
| O($^3$P) + surface | $\rightarrow$ | products | (R18) |
| NO$_2$ + O($^3$P) | $\rightarrow$ | NO + O$_2$ | (R19) |
| ISOP-NIT + O$_3$ | $\rightarrow$ | products + NO$_2$ | (R20) |
| ISOP-NIT + O($^3$P) | $\rightarrow$ | products + NO$_2$ | (R21) |
| ISOP + O$_3$ | $\rightarrow$ | products | (R22) |
| ISOP + O($^3$P) | $\rightarrow$ | products | (R23) |

The contribution (in addition to the thermal dissociation of ISOP-NIT) to NO$_2$ formation via reactions (R20) and (R21) in TDI-2 were assessed via numerical simulation (FACSIMILE/CHEKMAT release H010 (Curtis and Sweetenham, 1987)). The rate coefficients for the most important reactions are listed in Tab. 1; the complete reaction scheme is listed in the supplementary information (S9). Reaction times in heated and unheated section of TDI-2 were calculated from the temperature, internal diameter of the quartz tube and the flow rate. The rate coefficients for the gas-phase reactions of isoprene (IUPAC, 2017) and 2-methyl-2-butene (Herron and Huie, 1973) with O($^3$P) and O$_3$ were used as surrogates for the reactions of ISOP-NIT for which data is not available. The temperature-dependent dissociation rate coefficient of n-propyl nitrate was used to account for NO$_2$ from the thermal dissociation of isoprene-derived nitrates (Morin and Bedjanian, 2017). Wall loss of O($^3$P) in the instrument was estimated to be 90 s$^{-1}$ using the method as described in Thieser et al. (2016) and implemented in the model run.

The initial conditions for the simulation were 1 ppbv ISOP-NIT, 10 ppbv isoprene and 5 ppbv NO$_2$ at a cavity pressure of 724 hPa and a temperature of 298 K. The results obtained are shown by black curves in Fig.7. For temperatures up to 575 K the simulated thermograms with and without O$_3$ are almost identical, whereas at higher temperatures, the amount of NO$_2$ exiting the inlet decreases because of its reaction with O($^3$P) (R18), in broad agreement with the experiments carried out using TDI-2 as shown in Fig. 3b. The model simulations show (Fig. 7), that almost no O($^3$P) is formed by the thermal dissociation of O$_3$ at the lower temperatures of region II. Only at higher temperatures, is a significant fraction of O$_3$ (27 % at 703 K) converted to O($^3$P) with the majority subsequently lost at the inlet walls. These calculations underline the observation that the low temperature formation of NO$_2$ from ISOP-NIT seen when using TDI-1 cannot be explained with known gas-phase chemistry.

In summary, the experimental observations and the numerical simulations indicate that the presence of O$_3$ is required in the inlet for TDI-1 and in the chamber for TDI-2 to generate NO$_2$ from isoprene-derived nitrates at temperatures less than 475 K. We have shown that the generation of NO$_2$ from alkyl-nitrates at low temperatures using TDI-1 requires that the organic nitrate has a double-bond and that, while gas-phase reactions of O($^3$P) are responsible for the loss of NO$_2$ at high temperatures, they are NOT responsible for the conversion of isoprene-derived nitrates at lower temperatures in neither of the TD-inlets. The presence of glass beads (large surface area) favours the formation of NO$_2$ from ISOP-NIT at low temperatures. Altogether, these observations indicate that a surface-catalysed reaction involving ozone is the process most likely to be responsible for the conversion of ISOP-NIT to NO$_2$ at temperatures below those required for the gas-phase thermolysis.

### 3.2.3 Surface catalysed reactions with ozone

Quartz tubes contain impurities and surface defects that can provide reactive sites (RS) at elevated temperatures and the surface catalysed chemistry of ozone on e.g. mineral silicates is well known (Bulanin et al., 1994; Hanisch and Crowley, 2003a,b; Usher et al., 2003). $O_3$ can be surface-catalytically converted to $O_2$ (R25) by the formation and loss of reactive, oxygenated surface sites (RSO) via (R23) and (R24).

$$RS + O_3 \quad\quad \rightarrow \quad\quad RSO + O_2 \quad\quad\quad\quad\quad\quad\quad\quad\quad\quad\quad\quad\quad (R23)$$

$$RSO + O_3 \quad\quad \rightarrow \quad\quad RS + 2\,O_2 \quad\quad\quad\quad\quad\quad\quad\quad\quad\quad\quad\quad\quad (R24)$$

$$2\,O_3 \quad\quad\quad \rightarrow \quad\quad 3\,O_2 \quad\quad\quad\quad\quad\quad\quad\quad\quad\quad\quad\quad\quad\quad\quad (R25)$$

In order to test for ozone loss in our setup, $O_3$ in synthetic air was passed through the TDIs. The $O_3$ mixing ratio prior to entering the inlets was measured continuously using the ozone monitor (UV absorption). The ozone exiting the TD-inlets was converted to $NO_2$ (by addition of 1 ppmv NO (R4) in a 1.5 m long PFA tubing with 0.5 inch OD and a residence time of 5.2 s) and then measured in the 409 nm cavities. Figure 8a shows the results of such an experiment for TDI-1. The concentration of ozone before entering the inlets was 20.5 ppbv (open dots) which was detected as 17.1 ppbv after passing through the inlet at 298 K (black squares, 11:15 to 11:55). This represents a conversion efficency of 0.83 which matches that expected when considering the reaction time, NO concentration and the rate coefficient ($k_4 = 1.9 \times 10^{-14}$ cm³ molecule$^{-1}$ s$^{-1}$, IUPAC, 2021).

At $\approx$ 12:00 LT, upon heating the inlet to 473 K the ozone exiting TDI-1 was depleted by up to 27 %, while heating to 573 K results in further loss (up to 40 %). An analogous experiment using TDI-2 (Fig. 8b) showed that while some $O_3$ was also lost (e.g. 4 % at 573 K) this was much less than in TDI-1. As the gas-phase thermal decomposition of $O_3$ is negligible under these conditions (0.6 % at 575 K, Fig. 7) the loss of $O_3$ when passing through the TD-inlets indicates that surface catalysed ozone decomposition takes place (R23 and R24), especially in TDI-1 where larger surface areas are available.

We now consider the possibility that the conversion of isoprene-derive nitrates to $NO_2$ can be catalysed by surfaces in the presence of $O_3$. We note that previous work has shown that the heterogeneous ozonolysis of alkenes on glass or other surfaces can be more efficient than its analogous, gas-phase process (Dubowski et al., 2004; Stokes et al., 2008; Ray et al., 2013) and now consider the possibility that RSO is the mediating reactive species in TDI-1 in our experiments.

$$RSO + \text{ISOP-NIT} \quad\quad \rightarrow \quad\quad NO_2 + \text{products} \quad\quad\quad\quad\quad\quad\quad\quad\quad\quad (R26)$$

We first examine the possible contributors to ISOP-NIT formed in the reaction of $NO_3$ with isoprene, in which the dominant initial step (in air) is a sequential 1,4 Addition of $NO_3$ and $O_2$ to form $\delta$- and $\beta$-peroxy radicals e.g. $O_2NOCH_2C(CH_3)=CHCH_2OO$ (Schwantes et al., 2015). In the presence of $NO_3$, $RO_2$ or $HO_2$, the peroxy radicals react further to form "first generation" isoprene nitrates which contain carbonyl, hydroperoxidic and alcoholic groups such as $O_2NOCH_2C(CH_3)=CHCHO$, $O_2NOCH_2C(CH_3)=CHCH_2OOH$ and $O_2NOCH_2C(CH_3)=CHCH_2OH$, respectively. Note that most of the known, first-generation organic nitrates retain a C=C double-bond.

A hypothetical ISOP-NIT degradation scheme involving the initial attachment of RS-O to the remaining double bond is given in Fig. 9. We consider only the fate of the most stable surface-adducts, i.e. tertiary radical in case of δ-products and secondary

radicals in case of β-products. Both radical-adducts will react with $O_2$ to form organic peroxy radicals which may undergo H-shifts (via five- or six-membered rings) resulting in formation of a radical with its unpaired electron in the direct vicinity of the nitrate functionality. Such unimolecular processes may become competitive to bimolecular reactions under atmospheric conditions (Møller et al., 2019). A possible fate of this radical is decomposition to form a carbonyl compound via $NO_2$
elimination (Hjorth et al., 1990; Berndt and Boge, 1995; Vereecken et al., 2021). For δ-products, the tertiary product may also eliminate $NO_2$ under the formation of an epoxide.

With TDI-2, the conversion of ISOP-NIT to $NO_2$ at low temperatures (region I) is only observed when $O_3$ is present in the chamber (Fig. 3a), but not when it is added only to the inlet (Fig. 6b) implying that the presence of $O_3$ as an oxidizing agent in the SCHARK is the main difference between these two reaction systems. This suggests that the ozonolysis of isoprene may
play an important role. As the reaction between isoprene and $O_3$ leads to the formation of OH and additional $HO_2$ (Zhang et al., 2002; Malkin et al., 2010; Cox et al., 2020), the fraction of $RO_2$ reacting with $HO_2$ in this system is enhanced when compared to the experiments in which $NO_3$ was generated from $N_2O_5$. The major product from the reaction between nitrated δ- and β-peroxy radicals with $HO_2$ are hydroperoxides, such as $O_2NOCH_2C(CH_3)=CHCH_2OOH$ (Schwantes et al., 2015). With the intention of assessing the effect of $O_3$ on hydroperoxide yields, further model calculations were performed using the
Framework for 0-D Atmospheric Modelling (F0AM) (Wolfe et al., 2016). The $NO_3$ isoprene oxidation scheme is still subject of current research and uncertain which is why the Master Chemical Mechanism (MCM, version 3.3.1, http://mcm.leeds.ac.uk/MCM) (Jenkin et al., 2015) as well as the Reduced Caltech Isoprene Mechanism Plus (RCIMP, version 5, https://data.caltech.edu/records/247) (Wennberg et al., 2018; Bates and Jacob, 2019) were applied. Figure 10 shows the calculated fraction of hydroperoxides to ISOP-NIT using both models and mixing 22 ppbv isoprene with either 10.8 ppbv $NO_2$
and 150 ppbv $O_3$ (runs 1 and 3) or 3 ppbv $N_2O_5$ (runs 2 and 4) to achieve $NO_3$ in a flow-through chamber with an exchange rate of $2.73 \times 10^{-4}$ s$^{-1}$. Generally, the MCM predicts higher hydroperoxide yields than RCIMP. A possible reason might be that MCM has higher rates of $HO_2$ reformation from reactions of isoprene-derived alkoxy radicals than RCIMP does. For that reason, MCM only predicts an increase in the hydroperoxide yield by a factor of 1.33 when the $NO_3$ source is changed from $N_2O_5$ to $NO_2 + O_3$ (model runs 1 and 2), whereas RCIMP predicts a factor of 2.56 (model runs 3 and 4). The calculations thus
broadly support the hypothesis that unsaturated, nitrated hydroperoxides are involved in the low temperture (surface catalysed) formation of $NO_2$ from ISOP-NIT. Indeed, hydroperoxides not only have high affinity for surfaces but also have a rather weak O-O bond with a dissociation energy of ≈ 45 kcal/mol (Bach and Schlegel, 2020) and several routes to surface catalysed elimination of $NO_2$ appear feasible. Decomposition of isoprene-derived hydroperoxides within the sampling line has been observed for other instruments (Rivera-Rios et al., 2014). A possible degradation mechanism for an isoprene-derived
hydroperoxide which is abundant from the $NO_3$ + isoprene system (Schwantes et al., 2015) is depicted in Fig.11. In this case, $O_2NOCH_2C(CH_3)=CHCH_2OOH$ coordinates via hydrogen bonds to the $SiO_2$ surface prior to cleavage of the O-O bond to form OH and an alkoxy radical. The latter may dissociate via unimolecular reactions to form closed-shell products under elimination of $NO_2$ (Wennberg et al., 2018; Vereecken et al., 2021).

### 3.4 Elimination of O₃ and surface catalysed conversion of ANs at low temperatures

Our findings clearly show that that the combination of heated quartz surfaces and ozone catalyse the decomposition of the unsaturated nitrates formed in the reaction between $NO_3$ and isoprene at temperatures below 473 K. While the exemplary, surface catalysed processes depicted in Fig.9 and Fig.11 fulfil the requirement of conversion of ISOP-NIT to $NO_2$ we stress that these are purely hypothetical and we cannot state with any certainty that they are the reactions responsible for our observations. While it would be highly interesting to investigate such processes using different (e.g. surface sensitive)

techniques this is clearly beyond the scope of this paper and of the experimental capabilities of this research project. Instead, we take the more pragmatic approach and indicate potential methods to eliminate such unwanted reactions when using TD-inlets.

In principal, as the process that convert ISOP-NIT to $NO_2$ are clearly surface catalysed (involving formation of RSO) their impact can be reduced by using a surface that does not support formation of RSO. We therefore tested a third TD-inlet (TDI-

3), consisting of 55 cm PFA tubing (0.375 in. OD) with a 20 cm heated section. PFA-tubing has been routinely used as TDI for measurement of e.g. PAN, as it is relatively unreactive to the peroxy radicals formed (Phillips et al., 2013). The C-F-bond of PFA is very strong and nonpolar which should reduce the formation of RSO as well as adsorption of ISOP-NIT to the surface. The performance of a TDI made of PFA (TDI-3) was examined by performing an experiment with the SCHARK analogous to the one shown in Fig. 2. The resulting time-series of $NO_2$, ΣPNs (using TDI-3 heated to 448 K) and ΣANs (using

TDI-2 heated to 648 K) are depicted in Fig.12. At first, $O_3$ (5 SLPM synthetic air passed over a Hg lamp) and $NO_2$ (200 sccm of 1 ppmv) in 25 SLPM dry synthetic air were constantly introduced into the SCHARK. After detectable amounts of $NO_3$ and $N_2O_5$ had been formed, isoprene (9.8 sccm of 46.5 ppmv) was added leading (as expected) to almost quantitative depletion of $NO_3$ and $N_2O_5$.

During the following 3 hours the signal in the ΣANs cavity (TDI-2) increased to ~1.2 ppbv, while the signal in the ΣPNs

channel (~ 40 pptv) was close to the detection limit. A CIMS measurement obtained during this experiment validates that PAN mixing ratios are lower than 50 pptv. This result already confirms that 1) the ANs derived from $NO_3$ + isoprene reaction are not detected at $T < 448$ K when PFA is used instead of quartz and 2) as expected, there is no significant generation of peroxy nitrates in these experiments which would be detectable with both ovens at T = 448 K (see Fig. S6 in the Supplement).

This result not only confirms that the previous detection of ISOP-NITs at low TD-inlet temperatures when using TDI-2 and

especially TDI-1 were caused by heterogeneous reactions at the quartz surface, but also provides a solution to the problem of the separate measurement of ΣANs and ΣPNs using TD-inlets.

We recall however, that the reason for adding glass beads to the inlet was to suppress recombination reactions of $NO_2$ with peroxy radicals by providing a surface to scavenge the latter. The use of a PFA tube rather than quartz will certainly exacerbate this effect, as the rate of loss of $RO_2$ to PFA surfaces is expected to be lower than on quartz (Wooldridge et al., 2010). This

implies that corrective procedures based on numerical simulation may be necessary in some environments as shown by Thieser et al. (2016) and this may limit the useful deployment of the method to regions where $NO_X$ levels are sufficiently low that

reaction of RO$_2$ with NO and NO$_2$ become insignificant. In addition, similar to the observations of Sobanski et al. (2016), ClNO$_2$ can interfere with the detection of ANs. The experiment shown in Fig. S7 reveals that ClNO$_2$ is detected with TDI-2 at 698 K, but not with TDI-3 at 448 K.

**4 Summary, Conclusions and implications for atmospheric measurements of ΣPNs and ΣANs**

We have shown that the detection of isoprene-derived organic nitrates via its thermal dissociation in quartz / glass inlets can (in the presence of O$_3$) be accompanied by undesirable side reactions which broaden the thermograms and thus impede the separation of PNs and ANs signals by sampling through TD-inlets at different temperatures as is commonly practised. While our experiments deal with the nitrates formed in the NO$_3$-initiated oxidation of isoprene, it is very likely that similar broadening

of thermograms would also occur with nitrates formed from the oxidation of other terpenoids, as some organic nitrates derived from nighttime oxidation of e.g. limonene still contain a double bond (Fry et al., 2011) and/or contain hydroperoxy groups. Specifically, we find that the presence of O$_3$ in either the quartz TD-inlet or in a Teflon simulation chamber results in the generation of NO$_2$ from isoprene-derived nitrates in TD-inlets made of quartz at temperatures less than 475 K and that this only occurs when the organic nitrate either has a double-bond or a hydroperoxy group (or both). The formation of NO$_2$ from

ISOP-NIT was accelerated in the presence of glass-beads which indicates that a surface-catalysed reaction involving ozone and reactive surface sites is the process most likely to be responsible both for the conversion of ISOP-NIT to NO$_2$ at low temperatures (375-475 K) and the conversion of HNO$_3$ to NO$_2$ at high temperatures (> 550 K). By avoiding the use of O$_3$ or using a non-quartz TD-inlet, we were able to show that the ISOP-NIT thermogram is entirely consistent with those of saturated alkyl nitrates.

We show that surface-catalysed reactions on quartz TD-inlets involving O$_3$ represent a potential source of bias in measurements of ΣANs and ΣPNs during field observations, especially when isoprene is abundant. For example, we previously reported results from two campaigns carried out using the TD-CRDS system described here on the same rural mountain site (Kleiner Feldberg) and season (but in different years). We found that the average, relative abundance of ΣPNs and ΣANs was quite different with ΣANs ≈ ΣPNs during the PARADE campaign in 2011 and ΣPNs > ΣANs during the NOTOMO campaign in

2015 (Sobanski et al., 2017). In 2011, the TD-inlet was a quartz tube (i.e. similar to TDI-2) (Thieser et al., 2016), whereas in 2015 the inlet contained glass beads (i.e. similar to TDI-1) (Sobanski et al., 2016). With our present understanding of the role of surfaces and O$_3$ in TD-inlets, we cannot rule out that the observations during 2015 were biased to lower values for ΣANs and higher values for ΣPNs, although, as discussed by (Sobanski et al., 2017) there are other, meteorological factors which would have contributed.

For the detection of PNs we have shown that (at the lower temperature required to thermally dissociate PNs to NO$_2$) surface catalytic effects that convert ANs (or other species) to NO$_2$ can be completely avoided by using a TD-inlet made of a non-reactive material like PFA (TDI-3). In this case, O$_3$ does not appear to have any impact.

When using a quartz TD-inlet for conversion of ANs + PNs to $NO_2$ at higher temperatures the surface reactions that shift thermograms to lower temperatures are of less significance as, in any case, the role of the TD-inlet is to convert all ANs and all PNs to $NO_2$. However, in order to avoid detection of $HNO_3$ or HONO other materials may be more suitable than quartz. Sapphire, commonly used in microwave discharge generated plasmas owing to its high purity and non-reactive surface, may represent a useful alternative. Under humid conditions, some of the observed interferences become negligible: No $HNO_3$ appears to interfere with the ANs measurement and the interference of ANs to the PNs measurement is reduced in TDI-2, but not in TDI-1.

This study emphasizes the importance of characterising thermal dissociation inlets under conditions which are similar to those found in the atmosphere. We recognise that the impact of surface catalytic processes will vary from one inlet to the next (even if made from the same material) and all quartz-TDIs will not necessarily exhibit the same degree of conversion of BVOC-derived ANs at low temperature. For this reason, thermograms should be measured using trace-gases that are abundant in the atmosphere and the effect of e.g. $O_3$ and water vapour should be thoroughly investigated.

*Data availability.* The data underlying the figures is available on request from the corresponding author.

*Author contributions.* PD conducted the experiments, analysed the data and wrote the manuscript. RD was responsible for the CIMS measurements. JS designed and built the SCHARK. JNC designed the experiments and together with JL contributed to the manuscript.

*Competing interests.* The authors declare that they have no conflict of interest.

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

 **Figures and Tables**

**Table 1:** Central reactions used for numerical simulation of the thermal dissociation of organic nitrates in TDI-2.

| Reaction | Rate constant | Reference |
|---|---|---|
| ISOP-NIT + M → RO + NO$_2$ | $3.16 \times 10^{15}$exp($-19676$/T) s$^{-1}$ | Morin and Bedjanian, 2017 |
| O$_3$ → O($^3$P) + O$_2$ | Pressure dependent (see text) | Peukert et al., 2013 |
| O($^3$P) + O$_2$ + M → O$_3$ + M | $6.0 \times 10^{-34}$(T/300)$^{-2.6}$ [M] cm$^3$ molecule$^{-1}$ s$^{-1}$ | IUPAC, 2021 |
| O($^3$P) + O$_3$ → 2O$_2$ | $8.0 \times 10^{-12}$exp($-2060$/T) cm$^3$ molecule$^{-1}$ s$^{-1}$ | IUPAC, 2021 |
| O($^3$P) + ISOP-NIT → products + NO$_2$ | $3.9 \times 10^{-12}$exp(680/T) cm$^3$ molecule$^{-1}$ s$^{-1}$ [1] | Herron and Huie, 1973 |
| O($^3$P) + ISOP → products | $3.5 \times 10^{-11}$ cm$^3$ molecule$^{-1}$ s$^{-1}$ (298 K) | Paulson et al., 1995 |
| O($^3$P) + wall → | 90 s$^{-1}$ | See text |
| O($^3$P) + NO$_2$ → NO + O$_2$ | $5.1 \times 10^{-12}$exp(198/T) cm$^3$ molecule$^{-1}$ s$^{-1}$ | IUPAC, 2021 |
| O$_3$ + ISOP → products | $1.05 \times 10^{-14}$exp($-2000$/T) cm$^3$ molecule$^{-1}$ s$^{-1}$ | IUPAC, 2021 |
| O$_3$ + ISOP-NIT → products + NO$_2$ | $1.05 \times 10^{-14}$exp($-2000$/T) cm$^3$ molecule$^{-1}$ s$^{-1}$ [2] | IUPAC, 2021 |

Notes: [1]Value for 2-methyl-2-butene used. [2]Value for isoprene used.

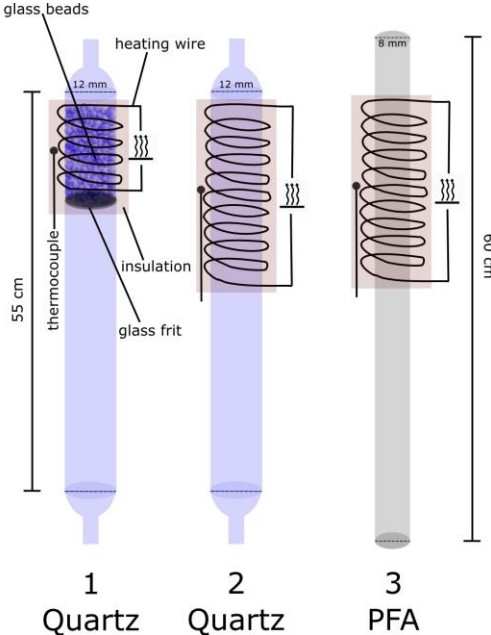

**Figure 1:** Schematic diagram of the thermal dissociation inlets TDI-1, TDI-2 and TDI-3.

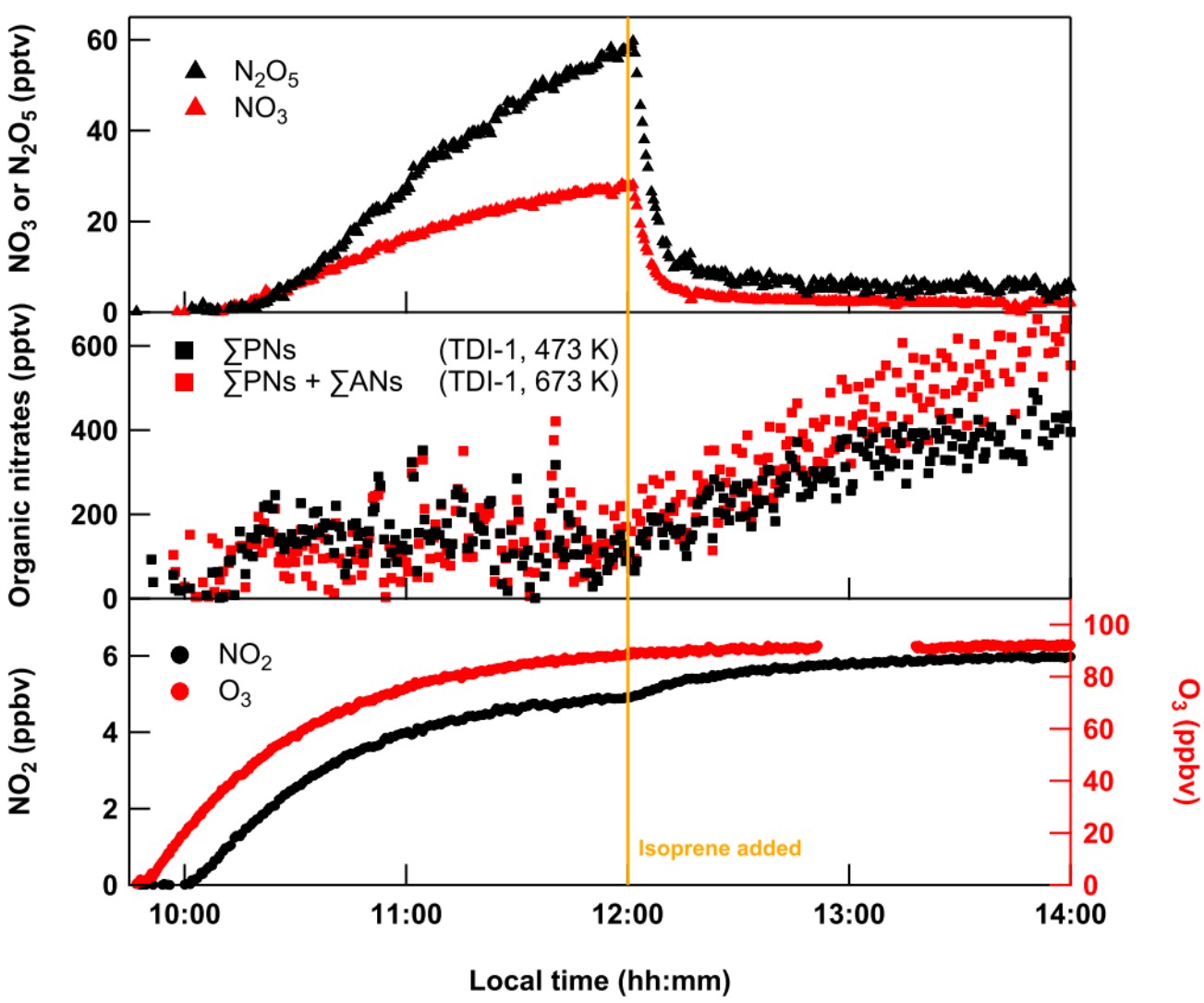


**Figure 2:** Evolution of the mixing ratios of $NO_2$, $\Sigma PNs$, $\Sigma PNs + \Sigma ANs$, $N_2O_5$, and $O_3$ when flowing 150 sccm $NO_2$ (from 1 ppmv bottle), 7 sccm isoprene (from a 46.5 ppmv cylinder) and 24 SLPM of zero-air (of which 5 SLPM were passed over a Penray lamp) into the SCHARK. Isoprene was added at 12:00 after the system was close to steady-state. Note that the $NO_2$ and $N_2O_5$ mixing ratios were subtracted from the organic nitrate signals (middle panel).

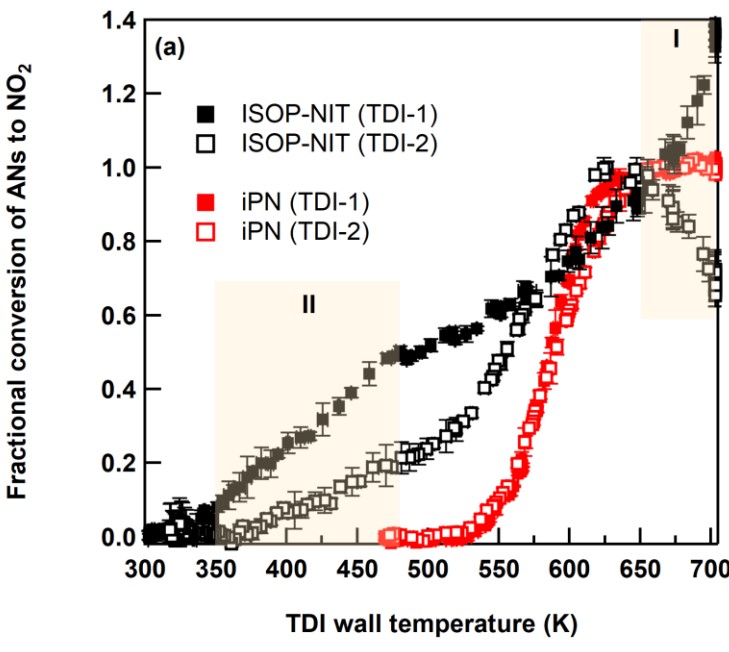

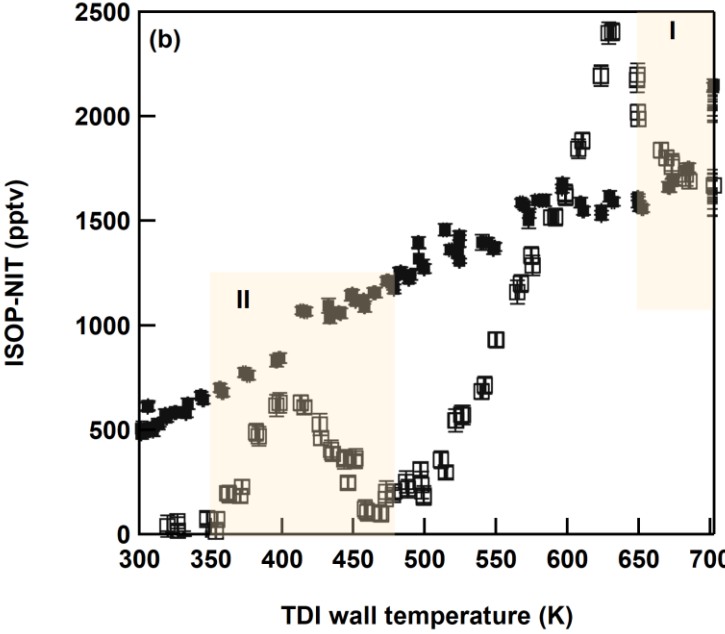

**Figure 3: (a)** Thermograms (relative to the signal at 650 K) of isoprene-derived organic nitrates (ISOP-NIT) and isopropyl nitrate (iPN) obtained with TDI-1 (solid symbols) and TDI-2 (open symbols) under dry conditions. **(b)** Absolute thermograms of ISOP-NIT obtained with TDI-1 (solid symbols) and TDI-2 (open symbols) under humid conditions (RH = 34 %, 22°C). Regions (I and II) with unexpected detection of $NO_2$ are shaded yellow. Error bars denote standard deviation ($1\sigma$, 30 s) of the signal.

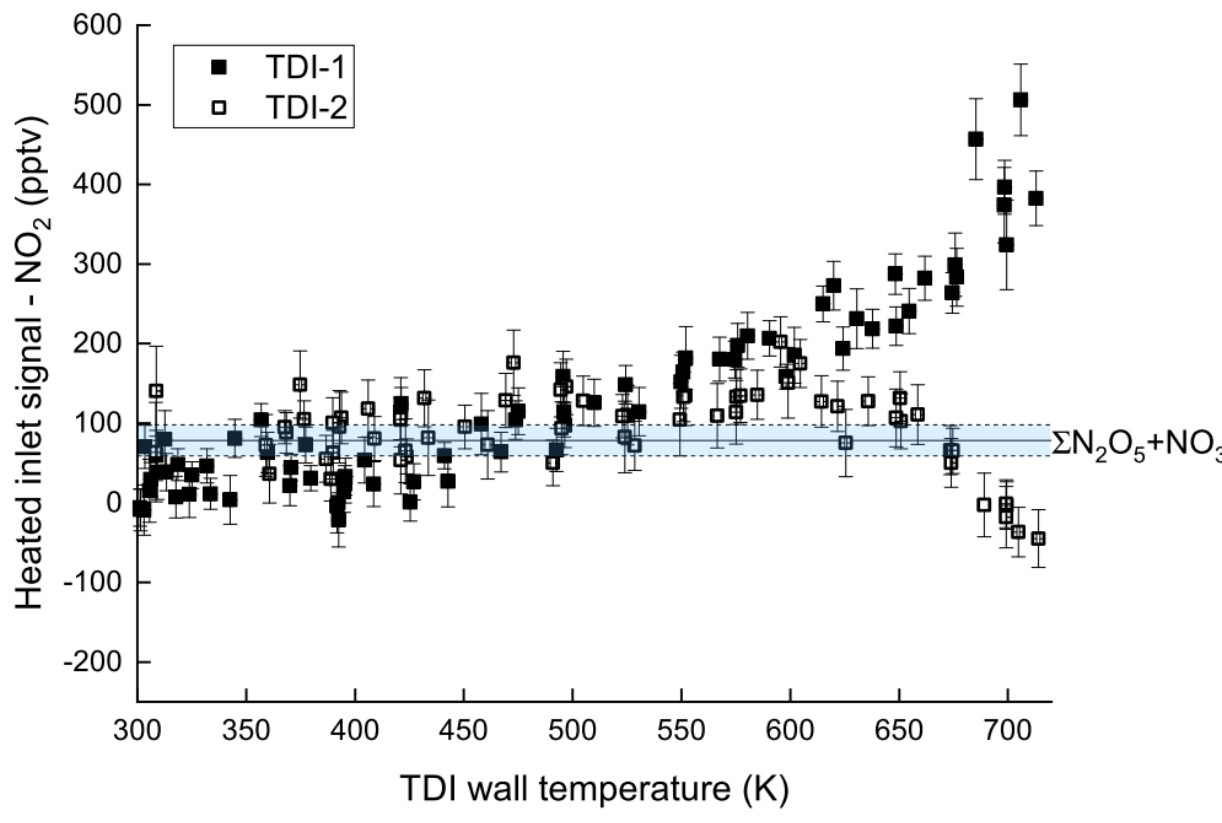

**Figure 4:** Temperature-dependent NO$_2$ detection when sampling 2.75 ppbv NO$_2$ and 146 ppbv O$_3$ in 23 SLPM dry synthetic air from the SCHARK through TDI-1 and TDI-2. The signal from the NO$_2$ cavity (no TDI) has been subtracted from both datasets. The mixing ratios of
N$_2$O$_5$ + NO$_3$ (with associated uncertainty) is indicated by the blue area.

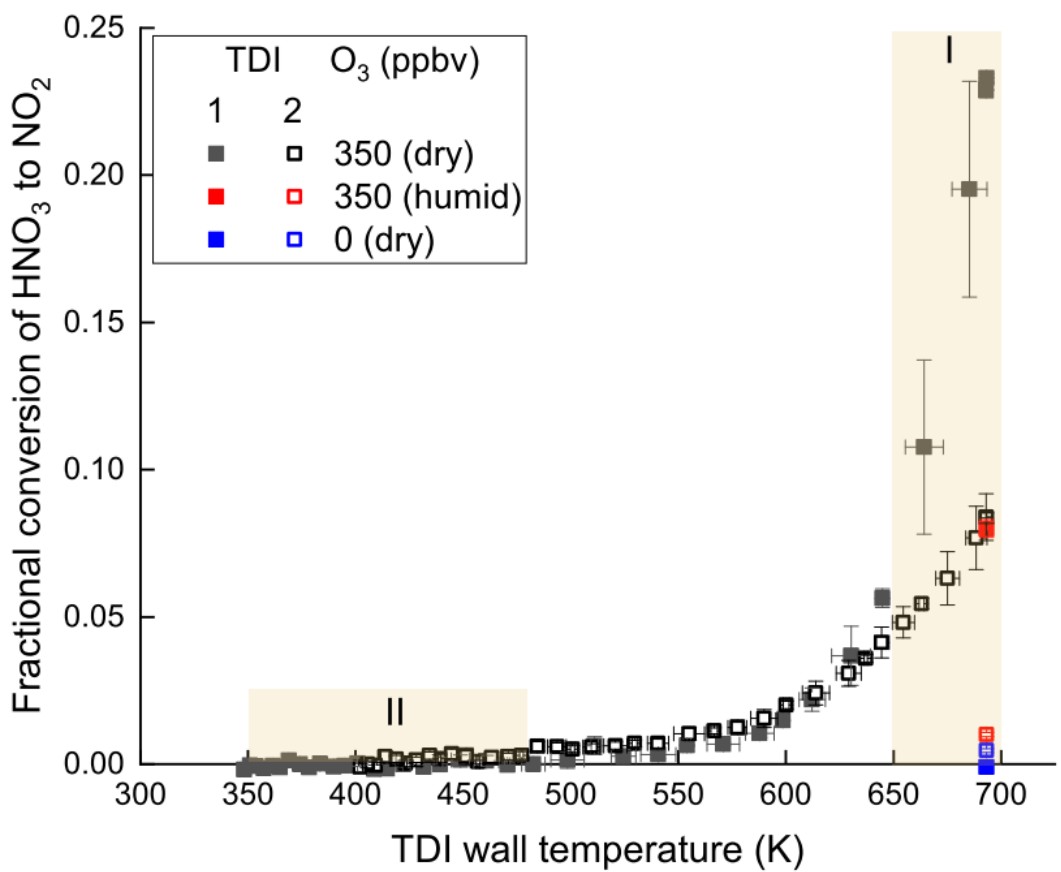

**Figure 5:** Thermograms of nitric acid (22 ppbv) in either dry (black) or humidified air (RH = 55 % at 23°C, red) obtained with TDI-1 (closed squares) and TDI-2 (open squares). The O₃ mixing ratio was either zero or 350 ppbv in the non-humidified experiment and 350 ppbv in the humidified experiment. The error bars denote standard deviation (1σ, 1 min) of the signal.

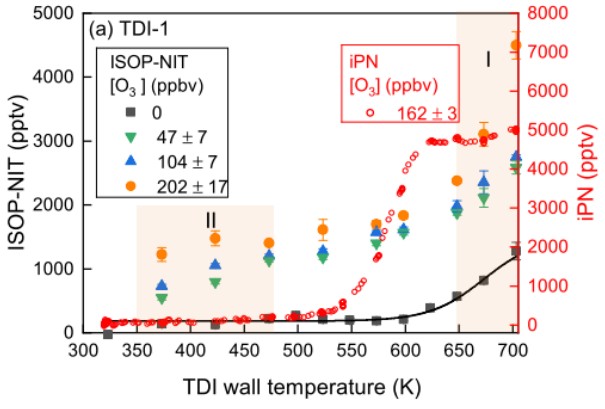

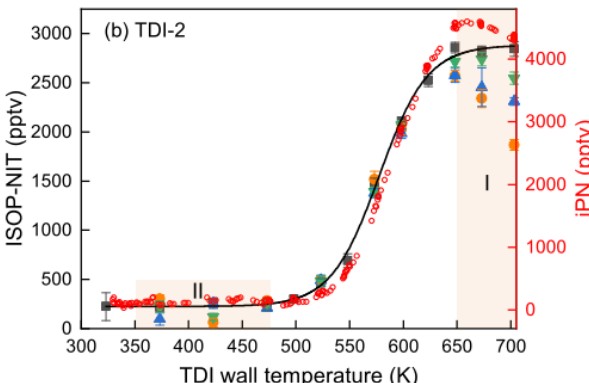


**Figure 6:** (a) Thermograms of isoprene-derived nitrates generated by mixing $N_2O_5$ (as $NO_3$ source) and ~ 20 ppbv isoprene in the SCHARK under dry conditions and sampling via TDI-1. The mixing ratio of $O_3$ (added only to the inlets) was varied from 0 to 202 ppbv. A thermogram of isopropyl nitrate (iPN) in the presence of 162 ppbv $O_3$ (and 3 ppbv $NO_2$ impurity) sampled through the same TDI is shown for comparison. The black solid line is a Boltzmann sigmoidal fit of the ozone free experiment. The error bars show the standard deviation of the signal (1$\sigma$).

(b) Same as (a) but using TDI-2.

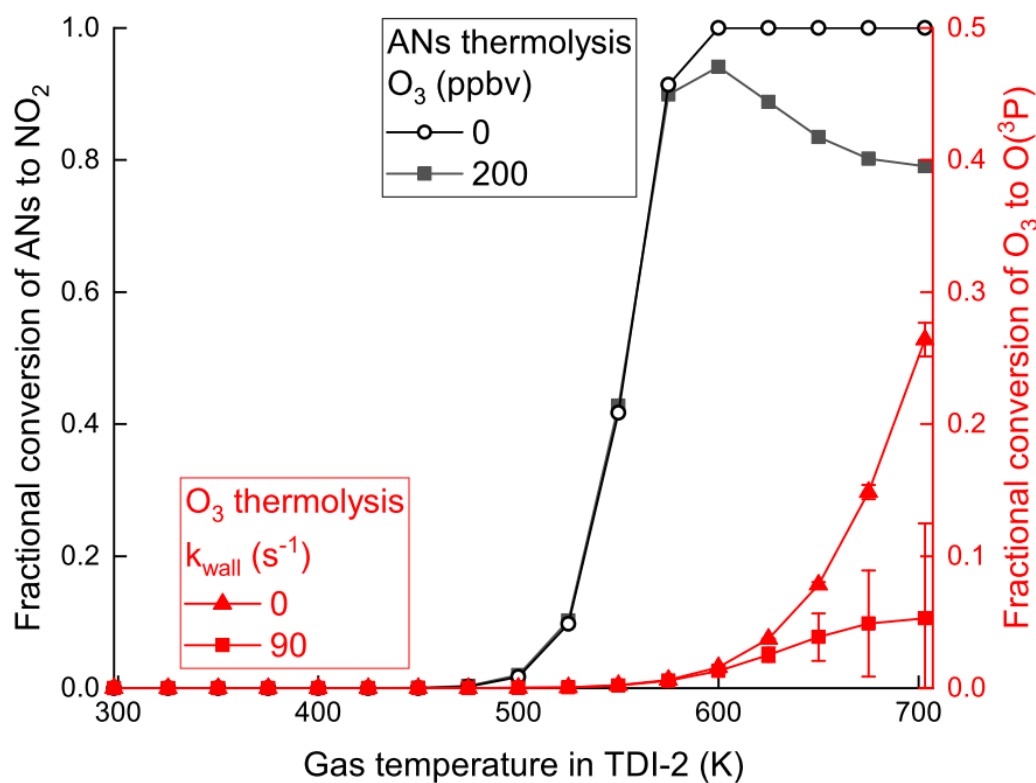

**Figure 7:** Black lines and data points: Simulated, fractional conversion of n-propyl nitrate to $NO_2$ for TDI-2 without and with $O_3$ (200 ppbv). Red lines and data points: Simulated fractional conversion of $O_3$ to $O(^3P)$ within the heated section of TDI-2 with and without wall loss of $O(^3P)$. Error bars denote the standard deviation ($1\sigma$) of the fractional conversion changing over time passed in the heated section.



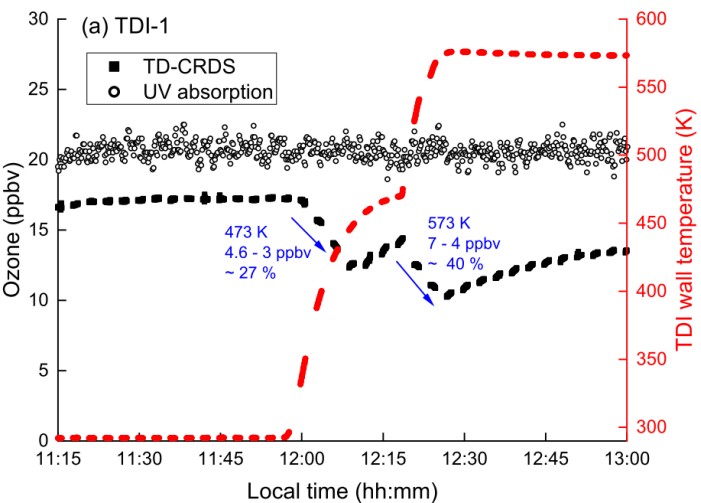

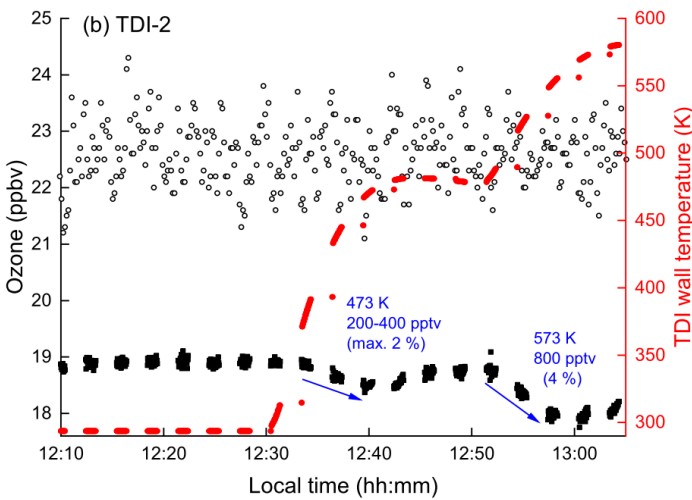


**Figure 8:** Time-series of O$_3$ mixing ratios before and after passing though TDI-1 (a) or TDI-2 (b).

**Figure 9:** Possible mechanism for surface-catalysed conversion of first generation ISOP-NIT in TDI-1. RS-O represents a reactive surface site.

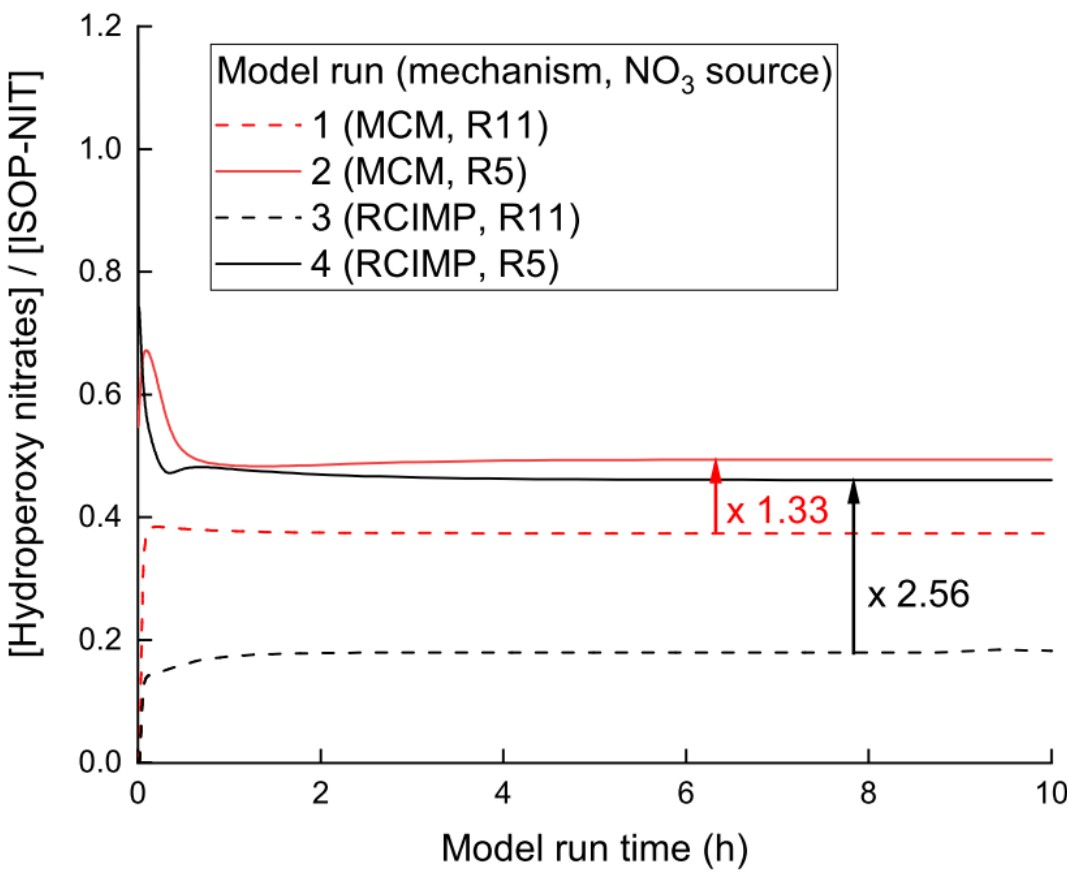

**Figure 10:** Simulated fractional contribution of hydroperoxy nitrates to ISOP-NIT using schemes taken from the MCM red) and from RCIMP (black). The simulations reproduce the experimental conditions in Fig.3 and Fig.6, with 22 ppbv of isoprene together with either 3 ppbv $N_2O_5$ as $NO_3$ source (runs 1 and 3) or 10.8 ppbv $NO_2$ + 150 ppbv $O_3$ (runs 2 and 4).



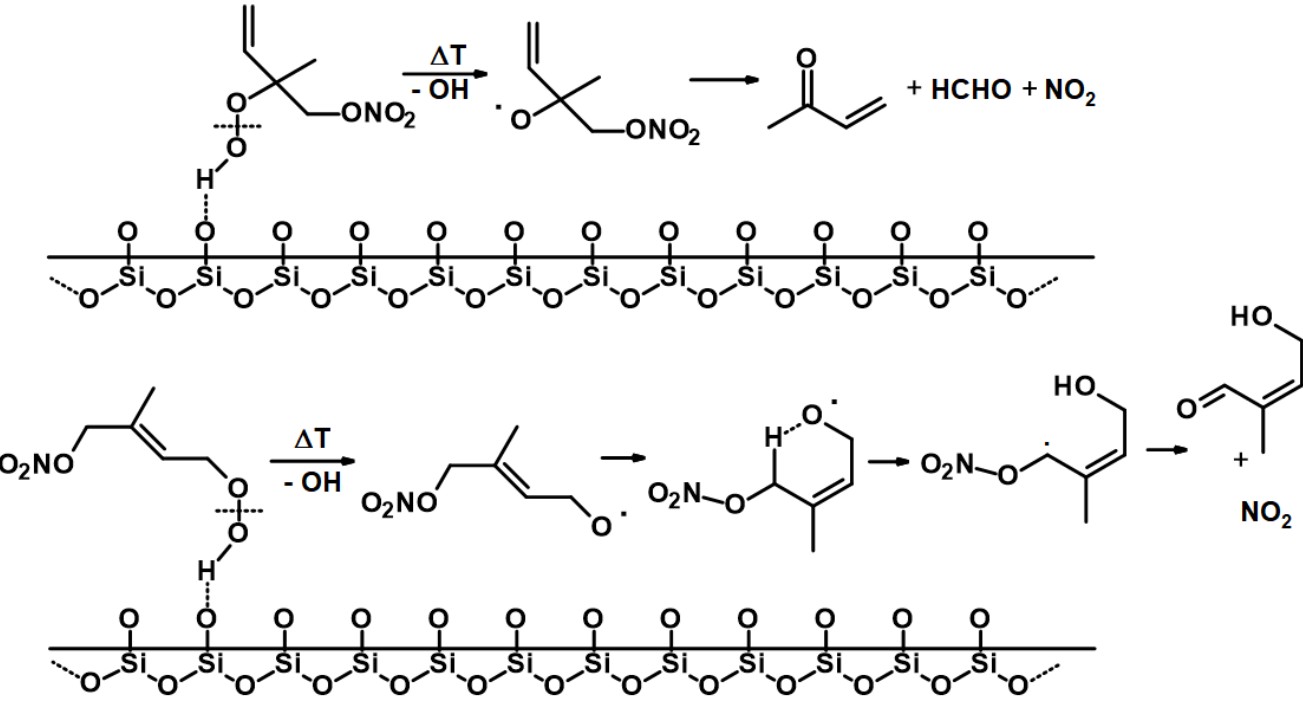


**Figure 11:** Potential degradation pathways of isoprene-derived hydroperoxides on a heated quartz surface.

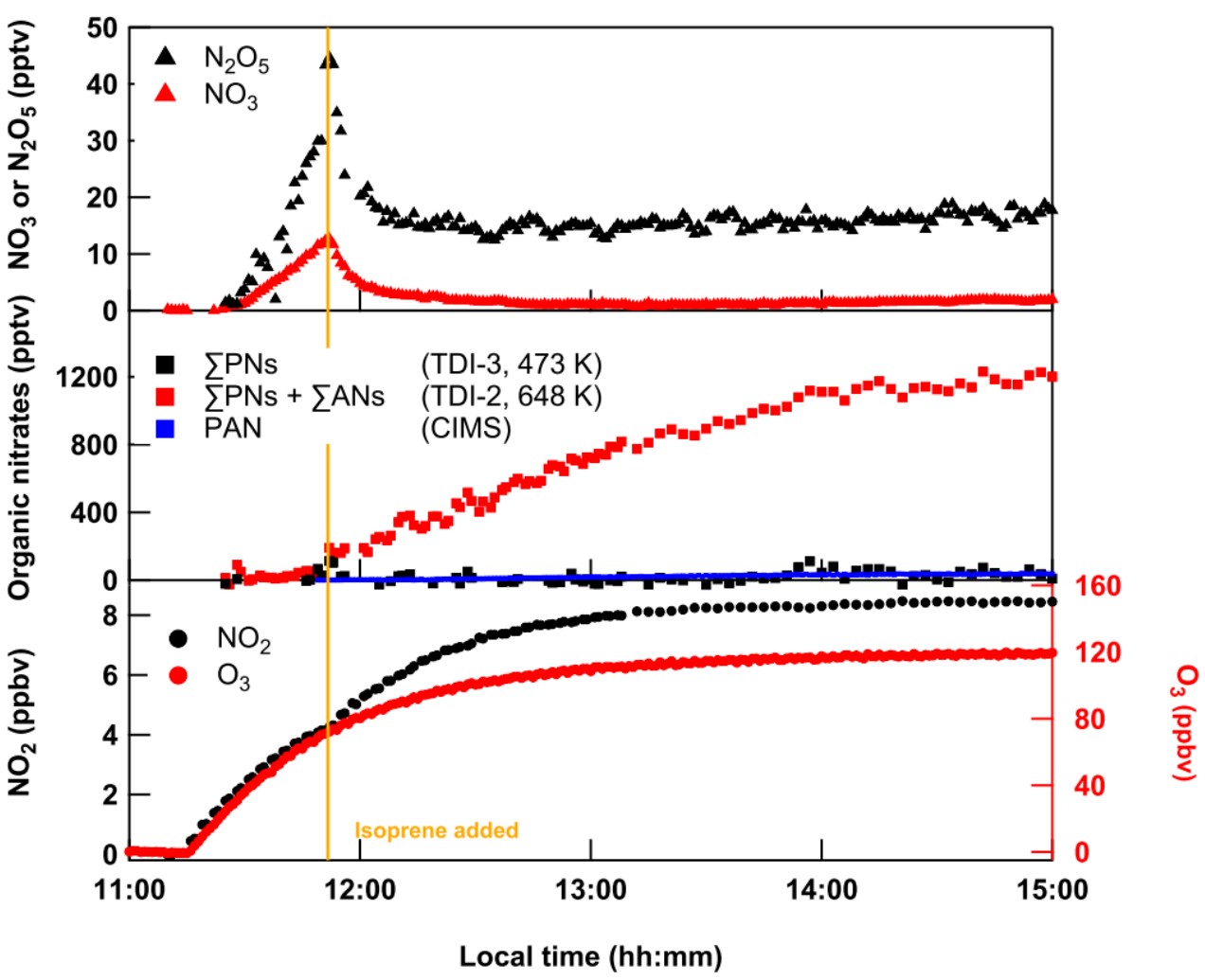

**Figure 12:** Mixing ratios of $NO_2$, $\Sigma PNs$ with TDI-3 and $\Sigma PNs$ + $\Sigma ANs$ with TDI-2 obtained from constant introduction of 200 sccm $NO_2$, and 9.8 sccm isoprene in 25 SLPM (of which 5 SLPM were passed over a Penray lamp to generate $O_3$) dry synthetic air into the SCHARK. A CIMS measurement of PAN is appended (blue squares).